# Application of Machine Learning in Predicting Osteogenic Differentiation of Mesenchymal Stem Cells

**DOI:** 10.3390/bioengineering12101089

**Published:** 2025-10-09

**Authors:** Hanyue Mao, Zheng Zhou, Ying Yang, Kunlu Lin, Chuyao Zhou, Xiaoyan Wang

**Affiliations:** Department of Biology and Chemistry, College of Science, National University of Defense Technology, Changsha 410073, China; maohanyue22@163.com (H.M.); zhengzhou23a@163.com (Z.Z.); yingyang015952@163.com (Y.Y.); kunlulin@163.com (K.L.); chuyaozhou.06@foxmail.com (C.Z.)

**Keywords:** mesenchymal stem cells, machine learning, osteogenic differentiation

## Abstract

This article reviews the progress made in applying machine learning to predict the osteogenic differentiation of mesenchymal stem cells. Bone defects pose a significant clinical challenge due to limitations of traditional therapies such as autologous bone graft donor shortages, allograft immune risks and so on. Mesenchymal stem cells offer a promising solution for bone regeneration due to their osteogenic differentiation potential, but their clinical utility is hindered by unpredictable differentiation efficiency and heterogeneity. Machine learning has emerged as a powerful tool to address these issues by enabling early, non-invasive prediction of osteogenic differentiation and high-throughput analysis of complex data like morphology and omics. This review systematically summarizes the application of ML in three key areas: early prediction using cellular morphology, omics data analysis for biomarker discovery, and drug/biomaterial screening for enhancing osteogenesis. We compare the performance of multiple ML models like ResNet-50, LASSO, and random forests and highlight their advantages and limitations. Additionally, we discuss challenges in data standardization and model interpretability, and propose future directions for translating ML into clinical practice. This review provides a comprehensive overview of how ML can revolutionize MSC-based bone regeneration by improving prediction accuracy and optimizing therapeutic strategies.

## 1. Introduction

A bone defect refers to the destruction of the structural integrity of bone tissue in response to external forces, infection, or tumors, resulting in loss of function. Repair of bone defects has been a medical challenge in clinical practice. And with aging, the number of patients with bone defects is increasing. However, open osteotomies increase the risk of infection, leading to complications and even disability, and there are also problems such as insufficient autologous bone donors and immune rejection of allogeneic bone, the quantity and quality of which cannot be guaranteed [1,2,3]. 3D printing and bioactive materials have also been used for bone repair, but the mechanical properties (e.g., modulus of elasticity) of artificial bone materials differ significantly from those of natural bone, and there is still room for optimization.

In contrast, stem cell therapy demonstrates significant potential in treating bone defects. Stem cells—particularly mesenchymal stem cells (MSCs)—possess the potential to differentiate into osteoblasts and chondrocytes, enabling direct participation in repairing tissue-deficient areas. For instance, MSCs have already been applied to treat epidermolysis bullosa [4]. Bone marrow-derived MSCs have been shown to completely heal large bone defects in mouse models in as little as 4 weeks, significantly reducing the recovery time required for conventional bone grafts [5]. It also possesses a relatively mature evaluation system and direction [6]. However, stem cell therapies also face numerous challenges. Although adult stem cells (ASCs) are widely available, their differentiation potential and quantity may vary significantly depending on donor age, health status, and cell isolation and culture conditions, limiting their efficacy and consistency in clinical applications [7]. Therefore, predicting the osteogenic differentiation direction of stem cells during the pre-transplantation culture process is crucial to achieve maximum therapeutic effects. Traditional methods for predicting the osteogenic differentiation potential of mesenchymal stem cells (MSCs) primarily rely on endpoint detection after in vitro osteogenic induction, such as alizarin red S staining to observe calcific nodule formation, alkaline phosphatase (ALP) activity assays, or expression analysis of osteogenesis-related genes (RUNX2, OSTERIX, OPN, etc.). While intuitive, these methods have significant limitations: First, they are predominantly destructive assays requiring termination of cell culture, precluding real-time dynamic monitoring of the differentiation process. Second, the timing of detection is empirically determined, making it easy to miss critical early changes and resulting in findings that lag behind the actual differentiation state. Furthermore, traditional methods overlook cellular heterogeneity; population-level detection masks differences among subpopulations, limiting predictive accuracy. More critically, the in vitro induction environment differs markedly from the in vivo bone regeneration microenvironment. In vitro osteogenic capacity may not accurately reflect in vivo osteogenic potential, causing prediction results to diverge from animal or clinical efficacy. Consequently, traditional methods face challenges in timeliness, precision, and clinical relevance, necessitating the development of novel predictive technologies that are non-invasive, highly sensitive, capable of early prediction, and integrate differences between in vivo and in vitro microenvironments. With the rapid advancement of machine learning, an increasing number of artificial intelligence technologies are being applied to predict stem cell osteogenic differentiation, leveraging their advantages of high efficiency, precision, and reliability.

Existing review studies indicate that strategies for predicting mesenchymal stem cell (MSC) osteogenic differentiation have shifted from traditional molecular biomarker detection toward multimodal approaches integrating morphology, metabolomics, and artificial intelligence. These advances offer novel insights for bone tissue engineering, cell therapy quality control, and personalized bone repair material screening [8]. However, most current reviews focus on single omics data types [9] or specific models (e.g., LASSO) [10]. This review addresses several critical gaps: First, it compares 10 ML models (ranging from classical LASSO to deep learning ResNet-50) across three key application domains; Second, it synthesizes the application of diverse omics data (transcriptomics, proteomics, metabolomics) to provide a holistic view of osteogenesis prediction; Third, it surveys recent advances in drug and biomaterial screening (Figure 1). By synthesizing recent research, comparing ML model performance, and offering actionable recommendations for standardization and interpretability, this paper fills this gap Figure 2.

## 2. Method

This systematic review rigorously adhered to the PRISMA guidelines to ensure methodological transparency and reproducibility in literature selection. Comprehensive searches were conducted across PubMed, Web of Science, and Scopus databases from January 2018 to June 2025 using a carefully designed combination of keywords: (“machine learning” OR “deep learning” OR “artificial intelligence”) AND (“mesenchymal stem cells” OR “MSCs”) AND (“osteogenic differentiation” OR “bone regeneration”) AND (“prediction” OR “screening”). The initial search yielded 1234 records, which were subsequently deduplicated using EndNote, resulting in 778 unique articles. Two independent reviewers then performed a rigorous title/abstract screening process with strict inclusion criteria: studies must employ machine learning techniques specifically for predicting MSC osteogenic differentiation, provide quantitative performance metrics (e.g., accuracy, AUC, sensitivity), and include detailed methodological descriptions of both the biological experiments and computational approaches. This screening phase excluded 566 articles primarily due to irrelevance to ML applications (*n* = 312), focus on non-osteogenic differentiation (*n* = 187), or being non-research articles (*n* = 67).

The remaining 212 articles underwent full-text evaluation against more stringent criteria, where an additional 154 studies were excluded for specific reasons: 35 were non-English publications, 10 represented duplicate publications from the same research groups, 45 involved inappropriate cell types (e.g., not MSCs), and 64 lacked essential ML model validation or proper performance reporting. The final 58 included studies were selected based on their scientific rigor, with particular emphasis on those demonstrating methodological innovation, clinical relevance (e.g., validation using patient-derived MSCs), or mechanistic insights (e.g., linking morphological features to molecular pathways). While these covered three primary research directions—morphology-based prediction, omics data integration, and biomaterials screening—the review’s focus on ML model performance comparison and mechanistic exploration led to more detailed discussion of 36 core references through reading the whole article, supplemented by 8 additional citations from recommended readings to provide broader context. This selective approach ensured the review maintained depth in analyzing key methodological advances while comprehensively covering the field’s current state (Figure 3).

## 3. Early Prediction Based on Morphology

In MSCs research, morphological feature extraction technology, as an important analysis tool, has far-reaching significance for the early prediction of cell differentiation process. While traditional biochemical marker assays take weeks and are limited to the late stage of differentiation, morphological analysis offers the possibility of early prediction by capturing the early dynamic changes of cells in real time through microscopic techniques [7]. Typically, morphological features of cells can be obtained through microscopy techniques such as phase-contrast microscopy, fluorescence microscopy, and laser confocal microscopy [11,12]. These techniques enable researchers to observe the morphological changes of cells in real time, and then extract information such as the size, shape, edge characteristics and internal distribution of cells. For example, during osteogenic differentiation, the morphology of BMSCs changes from spindle shape to polygonal shape, and the cell volume increases, and the arrangement of the cytoskeleton changes [13].

In recent years, with the continuous development of machine learning and image processing technologies, morphological feature extraction techniques have gradually moved toward high-throughput and automation. Using deep learning models, people can automatically identify and extract different morphological features from large-scale cell images. Among them, the CNN algorithm performs well in image recognition. M. Mai’s research team developed and compared four convolutional neural network (CNN) models (VGG 19, InceptionV3, ResNet 18, and ResNet 50), combining live-cell imaging technology with deep learning techniques to achieve efficient prediction of adipogenic and osteogenic differentiation in hMSCs. The ResNet-50 model outperformed other models in terms of area under the curve (AUC > 0.96) and accuracy, This is attributed to its residual block design, which effectively mitigates the “vanishing gradient” problem in deep networks, making it suitable for high-resolution cell image analysis (e.g., 1024 × 1024 pixels) [14]. In contrast, VGG19 (with 16 convolutional layers) tends to overfit on small datasets (e.g., <1000 images) due to its overly deep architecture, and its computational cost is twice that of ResNet-50; Inception V3’s multi-scale feature extraction (e.g., 1 × 1, 3 × 3 convolution kernels) performs poorly in classifying cell morphology (e.g., polygonal, spindle-shaped) (AUC = 0.89) [14]. Shi also used the CNN model by obtaining bright-field images taken at different time points (e.g., days 0, 1, 3, 5, and 7) to perform quantitative analysis of MSC morphological features, finding that morphological features such as cell area and edge roughness are significantly correlated with osteogenic differentiation potential, which were then input into the ResNet-50 model. The results showed that morphological features from just 24 h (day 1) could achieve a 96.3% accuracy rate in predicting osteogenic differentiation, significantly outperforming traditional methods (such as ALP activity detection, accuracy rate = 78.5%) [15]. However, ResNet-50 requires large training datasets (≥10,000 images) and is sensitive to imaging parameters such as resolution. In experiments simulating real-world label noise, ResNet-50 saw its accuracy drop by approximately 8–10% under 20% label noise. Although it outperformed VGG16 and ANN, its performance remained significantly lower than on clean data [16]. Vision Transformers (ViTs) offer a promising alternative. Their self-attention mechanism captures global cellular features more effectively than CNNs. ViTs have been widely applied in medical image recognition [17]. Mai also mentioned exploring this direction as a potential next step in research [14]. The aforementioned studies demonstrate that deep learning models, particularly ResNet-50, exhibit outstanding performance in morphological prediction. However, their dependence on data volume and quality, coupled with the potential of ViTs, suggests that future efforts should focus on further optimizing models to accommodate a broader range of application scenarios.

In addition to using image recognition, morphological features can also be converted into parameters for analysis. H. Sasaki et al. used LASSO to select morphological features of MSCs (such as cell spreading area and shape factor) and retained key features related to osteogenic differentiation (such as cell area > 1000 μm^2^), constructing a predictive model. This enabled the early prediction of the differentiation potential of MSCs for osteogenic, adipogenic and chondrogenic differentiation [8]. LASSO feature selection reduced overfitting, achieving an accuracy rate of 82% in early prediction, and L1 regularization effectively reduced overfitting, making it suitable for small sample data. These non-invasive methods avoid the damage to cells caused by traditional detection methods while enabling large-scale cell screening through high-throughput image processing. In addition to direct image analysis using a microscope, morphological feature extraction techniques can be combined with traditional biochemical labeling methods. By comparing the relationship between morphological features and biochemical markers, the predictive capability of MSC osteogenic differentiation processes can be further enhanced. Studies have shown that certain morphological features, such as cell area and cell shape factor, exhibit significant correlations with the expression levels of osteogenic markers. In Matsuoka’s research, ALP activity measured on day 14 of MSC culture and calcium deposition rate measured on day 21 were used as osteogenic differentiation biomarkers, and ridge regression algorithms were employed to assess the osteogenic differentiation potential of mesenchymal stem cells [18]. By using a ridge regression model and optimizing image acquisition and analysis methods, and with alkaline phosphatase activity and calcium deposition as biological markers, researchers found that morphological features obtained just 3 days before cell differentiation could accurately predict osteogenic differentiation outcomes after 3 weeks [19]. This links ML to tangible therapeutic outcomes. Zeng developed a convolutional neural network (CNN) model based on dynamic images of live cells, successfully predicting differences in osteogenic differentiation efficiency among MSCs from different donor sources by tracking cell migration trajectories and pseudopod formation patterns, thereby revealing the deep connection between morphology and cell fate [20]. Through parametric analysis and multimodal fusion, the integration of morphological features with biochemical markers significantly enhances prediction accuracy and early detection capabilities, providing more reliable evidence for clinical application.

The complexity and diversity of morphological data simultaneously drive innovation in machine learning algorithms. Traditional image analysis methods depend heavily on manually crafted feature extraction and are susceptible to subjective factors. Deep learning, however, automatically extracts key features through end-to-end learning, which significantly improves the objectivity and accuracy of the analysis. For example, by integrating phase-contrast microscope images with a migration learning framework, a generalized prediction model across experimental platforms can be established, which effectively overcomes the bias caused by different microscope imaging parameters [21]. Or, by using a generative adversarial network (GAN), the cellular image data is enhanced, which solves the model overfitting problem in small-sample scenarios, ensuring that the prediction accuracy is still preserved over 85% under limited data conditions [22]. Additionally, multimodal fusion technology further expands the application boundary of morphological analysis. By integrating cellular morphological features with transcriptomic data and utilizing graph neural network (GNN) to construct a “morphology–gene” sociation map, COL1A1 and other osteogenic marker genes that are highly correlated with the spreading area of cells were identified, which provided a new perspective for understanding the molecular basis of morphological changes [20]. In summary, the application of deep learning in the prediction of osteogenic differentiation not only improves the timeliness and accuracy of prediction, but also promotes the updating of related algorithms, which has a broad application prospect and research value.

## 4. Analyzing Omics Data to Find Predictive Targets

With the rapid advancement of regenerative medicine, research on the osteogenic differentiation of mesenchymal stem cells (MSCs) has gained significant attention due to their central role in bone regeneration and disease treatment. Traditional methods rely on biochemical markers such as alkaline phosphatase (ALP) activity or calcium nodule staining to assess differentiation status, but these methods typically require several weeks and only provide results in the later stages of differentiation, making them unsuitable for early prediction or dynamic regulation [13]. In recent years, breakthroughs in high-throughput omics technologies have provided multi-dimensional data sources for elucidating osteogenic differentiation mechanisms. Machine learning (ML), as a powerful data analysis tool, integrates information from multiple levels such as genomics, transcriptomics, proteomics, and metabolomics, significantly improving predictive accuracy and research efficiency. This has made it possible to rapidly identify key genes and molecules predictive of osteogenic differentiation.

For example, single-cell RNA sequencing (scRNA-seq) combined with deep learning algorithms can reveal the heterogeneous characteristics of early-stage differentiated cells, completely transforming the study of mesenchymal stem cell heterogeneity. MSCs from different donor sources exhibit significant differences at the transcriptional and epigenetic levels, and this heterogeneity not only directly affects their osteogenic potential but also helps reveal their differentiation direction. Machine learning models can screen for core gene modules through clustering analysis and principal component reduction (PCA) techniques to construct gene regulatory networks [23]. By comparing the results of different machine learning algorithms in analyzing MSC transcriptome data, Zhou found that the KNN algorithm achieved the highest overall accuracy (90.63%) in predicting MSC osteogenic differentiation, providing a straightforward method for forecasting the differentiation direction of MSCs [24]. For example, Shen et al. used Seurat for data preprocessing (normalization and dimensionality reduction via PCA and UMAP) and the Leiden algorithm for clustering to identify MSC subpopulations with different osteogenic potentials. They then applied a random walk algorithm on the protein–protein interaction (PPI) network to screen for genes with similar topological structures to bone-specific genes (e.g., FOXA1) and validated their roles in regulating osteogenesis [25]. Additionally, recent studies have begun incorporating single-cell latent variable models (scLVM, f-scLVM) into MSC osteogenic differentiation prediction: Huang et al. utilized scATAC-seq combined with scRNA-seq to construct a “chromatin potential” latent variable. Using LightGBM, they predicted 7-day calcium deposition within 24 h, achieving an average correlation coefficient of r = 0.82 and an AUC of 0.91, enabling donor-level potential ranking. AUC = 0.91, enabling donor-level potential ranking [26]. Epigenetic modifications (such as DNA methylation) also play a key role in differentiation regulation. J. Chen analyzed methylation data from high-fat diet-induced aged mesenchymal stem cells using logistic regression (LR) and gradient boosting machine (GBM) models, identifying epigenetic silencing of the vitamin D receptor (VDR) as a key driver of reduced osteogenic potential. This model achieved an accuracy of 82% in predicting differentiation efficiency, with VDR methylation status emerging as a powerful biomarker [27,28]. K. Kamimoto utilized network inference models to identify key signaling nodes in cell identity determination, providing a theoretical basis for targeted interventions in osteogenic differentiation [29]. ZNF521 is a transcriptional repressor that restrains osteogenic differentiation by silencing RUNX2 and Osterix. Its expression level can serve as a negative biomarker of osteogenic potential [30]. Incorporating ZNF521 mRNA, promoter methylation or chromatin accessibility into machine-learning models will refine early prediction of bone-forming capacity and guide personalized regenerative strategies. By integrating omics data and machine learning methods, researchers can more precisely identify key genes and epigenetic markers regulating osteogenic differentiation, offering new insights for personalized treatment.

In addition to transcriptomics, dynamic analysis of other omics data such as proteomics, metabolomics, and spatial omics provides another dimension for understanding the mechanisms of osteogenic differentiation. Proteomics serves as a bridge between gene expression and function, and integrating transcriptomics and proteomics data using ML models can capture the time-delayed effects between gene expression and protein activity. By integrating transcriptomic and proteomic data, Kong et al. used a cross-modal Transformer model to fuse RNA-seq data from differentiating mesenchymal stem cells with proteomics data based on tandem mass tagging (TMT). The model identified a 24 h delay between ALP gene expression and protein activity, highlighting the importance of post-transcriptional regulation in osteogenesis [21]. Feng identified 205 differentially expressed proteins enriched in the extracellular matrix (ECM) pathway by analyzing proteomics data from osteoporosis patients using support vector machines (SVM) and random forest analysis. The ECM receptor interaction pathway was found to be a central regulator of osteogenic differentiation, providing targets for therapeutic interventions [28]. Metabolomics captures dynamic changes in energy metabolism (e.g., glycolysis, oxidative phosphorylation) during osteogenic differentiation. Machine learning (ML) models are used to link metabolite levels with differentiation efficiency. For example, Klontzas et al. analyzed metabolomics data from 2D and 3D mesenchymal stem cell cultures using a random forest model, identifying lactate and ATP as early predictors of osteogenic differentiation. The model achieved an accuracy of 89% in predicting differentiation outcomes, demonstrating a positive correlation between lactate levels and ALP activity [9]. Spatial omics technologies (e.g., spatial transcriptomics, proteomics imaging) map the spatial distribution of cells in the bone marrow microenvironment, revealing how microenvironmental factors (e.g., hypoxia, mechanical stress) regulate differentiation [31]. Piña et al. utilized the Visium platform combined with single-cell data to construct the first spatial transcriptomic map of mouse embryonic palatal fusion. They discovered that osteogenesis-related genes (such as Runx2, Spp1, and Sost) were significantly upregulated only after fusion and were precisely localized to the ossification centers. Further validation revealed that Deup1 and Lrrc23 are not only highly expressed in nasal epithelium but also colocalize within palatal mesenchyme, suggesting their potential role in regulating osteogenic differentiation [32]. Bandyopadhyay et al. used spatial transcriptomics and multi-beam ion imaging (MIBI) to map MSC subpopulations in human bone marrow, finding that osteogenic precursors aggregate near blood vessels. They applied a graph convolutional network (GCN) to simulate spatial interactions between MSCs and their microenvironment, predicting that hypoxia (via HIF-1α) enhances osteogenic differentiation [31]. The dynamic integration and spatial analysis of multi-omics data provide novel insights into the microscopic mechanisms of osteogenic differentiation and lay the foundation for developing targeted therapeutic strategies.

Machine learning, with its strong ability to process high-throughput data, is uniquely suited for omics data processing, and its combination with a variety of omics data has successfully achieved the function of accurately and rapidly predicting the differentiation of mesenchymal stem cells. While omics data provide rich molecular insights, challenges remain in data integration (e.g., dealing with different data types and scales) and model interpretability (e.g., understanding how ML models identify predictive targets). Future research should focus on developing multi-omics integration frameworks (e.g., multi-modal transformers) and interpretable ML model to improve the translational value of omics-based predictions. Additionally, standardizing omics data acquisition will enhance the reproducibility of results.

## 5. Drug and Biomaterial Selection

In drug and biomaterial selection for osteogenic differentiation of MSCs, machine learning has demonstrated the potential for revolutionary applications, significantly accelerating traditional experimental processes and improving screening efficiency through data-driven high-throughput analysis and predictive model optimization. In the field of drug selection, machine learning is able to rapidly identify candidate compounds with osteogenic induction potential and analyze their molecular mechanisms by integrating multi-dimensional data from the transcriptome, proteome and metabolome. For example, vitamin D analogs (e.g., Calcitriol) significantly increase the expression levels of osteogenic markers like Alkaline Phosphatase and osteocalcin by binding to the VDR. Through multi-omics analysis, VDR was found to serve a central regulatory function in the aging process of MSCs, and its expression level was negatively correlated with the risk of osteoporosis, which was used as a molecular target to build a machine learning model for the screening of anti-osteoporosis drugs in Chen’s study [27]. Shengxue Busui Decoction (SBD) shows efficacy in osteonecrosis clinics, but its mechanism is unknown. Machine learning combined with experimental validation reveals that SBD inhibits osteoblast apoptosis by regulating the PI3K/Akt and VEGF pathways, proving the efficacy of SBD [33]. Hitora utilizes machine learning techniques to screen for osteoclast differentiation inhibitors from natural products. While primarily targeting osteoclasts, its methods (e.g., data-driven compound activity prediction) can be directly migrated to osteogenic differentiation drug screening, and are particularly Y-referenced for high-throughput data processing and candidate compound prioritization [34]. Other researchers have used high-throughput screening techniques to rapidly and cost-effectively identify osteogenic compounds including apigenin, baicalein, and T63 [35,36,37,38], demonstrating that machine learning can significantly improve drug screening efficiency. In exception to this, reinforcement learning algorithms are able to accurately predict drug toxicity thresholds and achieve drug dose optimization by simulating the dynamic relationship between drug concentration gradients and cellular response. For example, the DRUG-seq technology developed by Ye combined with machine learning enabled rapid analysis of single-cell transcriptome data to quantify the intensity of regulation of osteogenesis-related genes such as RUNX2 and BMP2 by different doses of strontium ranelate, providing a theoretical basis for the personalized design of clinical drug regimens [39]. These studies show the great potential of machine learning in drug screening.

In the field of biomaterial design, machine learning has significantly improved the efficiency of developing bone repair materials by correlating the physicochemical parameters of the materials (e.g., stiffness, surface morphology, pore structure) with the cellular behavior data. Nanofiber scaffolds have become an ideal carrier to promote osteogenic differentiation of MSCs due to their structural properties that mimic the extracellular matrix. Zhang [22] analyzed the effect of nanofiber diameter and alignment density on osteogenic differentiation using a random forest model and found that disordered aligned fibers with diameters in the range of 200–400 nm could maximally promote calcaneal nodule formation, and the mechanism may be related to cytoskeletal remodeling and mechanical signaling (e.g., YAP/TAZ pathway). Some researchers have also used machine learning tools (TensorFlow and ArcGIS) to perform morphometric analyses and demonstrated that 30 nm hydroxyapatite nanoparticles alone or in combination with laser photobiomodulation significantly promote the proliferation and osteogenic differentiation of human umbilical cord mesenchymal stem cells (hUC-MSCs), with superior results to conventional growth factors [40]. Spinodal decomposition is a phase separation phenomenon in a thermodynamic process, resulting in the formation of bicontinuous structures with uniform feature sizes and smooth interfaces. Such structures are widely used in biomaterials, such as orthopedic implants in particular. Traditional methods to generate such structures are computationally expensive. It has been proposed to replace the traditional physical and mathematical approximation models with CNNs, combining the advantages of both, which is both flexible and efficient [41]. Machine learning can also be used to build biomaterial databases by integrating access to research. There are studies that analyze the design and performance of osteoinductive biomaterials through machine learning to address the challenges of insufficient data quality (small samples, high missing rates, and high-dimensional sparsity) and systematically integrate osteoinductive material databases [42]. The modulation of material stiffness is also crucial in determining cell fate, and a prediction model constructed by Liu [43] based on support vector (SVM) was able to predict the tendency of MSCs to differentiate towards osteogenic or adipogenic differentiation with 89% accuracy by inputting the elastic modulus of the material (1–100 kPa) and surface roughness data. In the design of functional drug-carrying scaffolds, to address the mismatch between slow-release kinetics and cellular response, M. E. Klontzas [9] proposed an optimization framework combining deep neural network (DNN) and metabolomics data to screen for the ratio of loaded drugs with the highest efficiency of osteogenic differentiation, by simulating the release profiles of BMP-2 from polylactic acid scaffolds and correlating the changes in metabolite dynamics (e.g., lactate dehydrogenase activity). In summary, machine learning can select biomaterials in a variety of ways and has broad applications in stem cell therapy.

## 6. Conclusions

Performance metrics for evaluating predictive algorithms include AUC (area under the ROC curve), which measures binary classification performance across thresholds, reflects overall correct classification accuracy, quantifies precision (the proportion of true positives among predicted positives) to assess the reliability of positive predictions, and quantifies key recall (the proportion of actual positives correctly detected) to capture the model’s ability to identify true positives. In the context of osteogenic differentiation studies, successful differentiation events are often rare. Relying solely on accuracy or AUC can lead to significant misinterpretation due to class imbalance. For example, a model achieving 90% accuracy by simply labeling all samples as negative may completely fail to detect osteoblasts. This necessitates analyzing the trade-off between precision and recall: high precision ensures the selected MSC subset is reliably distinguished (minimizing false positives) but may miss true positives (low recall), while high recall maximizes osteoblast detection (minimizing false negatives) at the cost of reduced precision. Therefore, complementary metrics like F1 scores and precision–recall curves are essential for robust evaluation in imbalanced scenarios. The major ML models discussed in this paper (such as CNNs, LASSO, and random forests) each have their own advantages and limitations in predicting MSC osteogenic differentiation, and these models are compared in detail below (Table 1).

Research findings indicate that machine learning significantly enhances the timeliness, accuracy, and scalability of predictions. Convolutional neural networks (CNNs) and transfer learning prove particularly effective for morphological data, while multimodal models demonstrate exceptional efficacy for omics integration.

## 7. Perspective

Although machine learning provides efficient and precise tools for predicting osteogenic differentiation, several challenges limit the translational value of current research. First, data standardization remains a major issue: inconsistent cell processing (e.g., isolation protocols, culture conditions) and imaging parameters (e.g., microscope settings, resolution) introduce bias into ML models, reducing their generalizability across experimental platforms. Second, the lack of model interpretability: most ML models operate as “black boxes,” making it difficult to understand how predictions are derived. This hinders their acceptance in clinical settings. Third, the absence of clinical validation: few studies test ML models on patient-derived MSCs or in vivo models, creating a gap between laboratory research and clinical application.

Future research should focus on addressing these challenges. Standardization efforts, such as establishing consensus protocols for cell processing and imaging, will enhance data quality and model generalizability. Researchers should prioritize interpretable ML models (e.g., SHAP, LIME, or graph neural networks with attention mechanisms) to reveal the biological mechanisms underlying predictions. Clinical translation requires testing models in patient populations, integrating clinical data (e.g., donor age, health status) with molecular and morphological data, and developing user-friendly tools for clinicians. Concurrently, machine learning necessitates improvements in data acquisition and quality, ensuring consistency in cell processing and imaging while minimizing human variability. Additionally, enhancing model transferability is crucial to maintain predictive performance across different experimental conditions or cell lines. In omics research, transcriptomic analysis remains predominant. Integrating additional omics data types (e.g., proteomics and methylation) could provide a more comprehensive assessment of cellular differentiation states. Moreover, predicting the osteogenic differentiation potential of MSCs from different sources represents another promising avenue for expansion. Future models could aim to elucidate such source-dependent variations in osteogenic differentiation (e.g., MSCs derived from Kilian polyps exhibit lower differentiation potential toward osteocytes and adipocytes compared to those from common nasal polyps [44]). Subsequent machine learning approaches should incorporate more granular analysis, specifically accounting for the origin of MSCs.

In summary, ML holds the potential to revolutionize the prediction of MSC osteogenic differentiation. However, realizing this potential requires continued efforts in standardization, interpretability, and clinical validation. By addressing these gaps, ML can become a powerful tool for optimizing stem cell therapies and advancing bone tissue engineering.

## Figures and Tables

**Figure 1 bioengineering-12-01089-f001:**
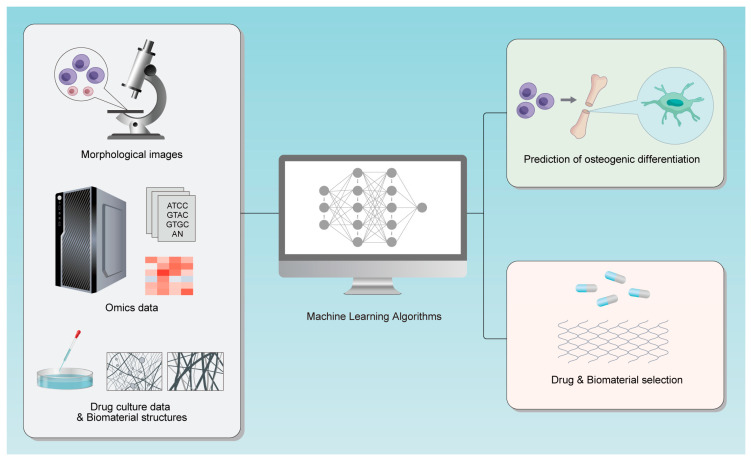
The research process diagram for this review.

**Figure 2 bioengineering-12-01089-f002:**
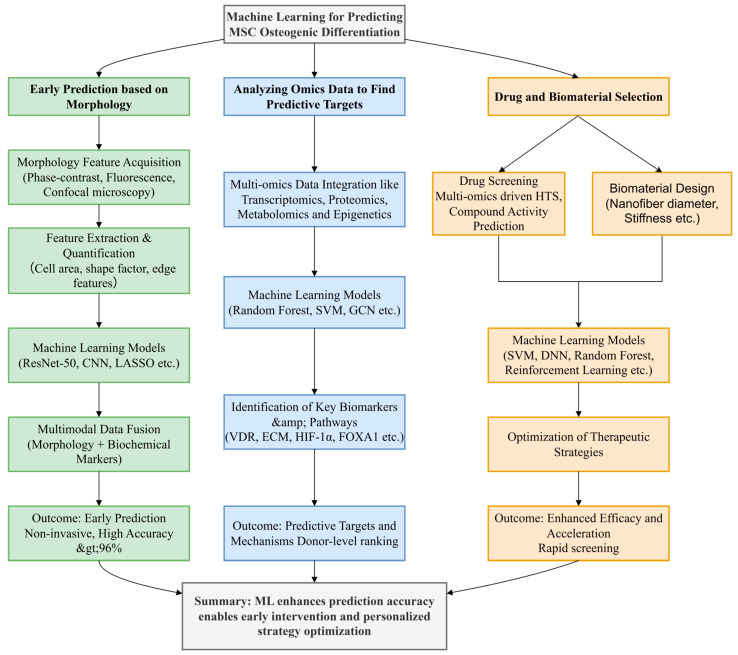
The analytical steps and structural points of this review.

**Figure 3 bioengineering-12-01089-f003:**
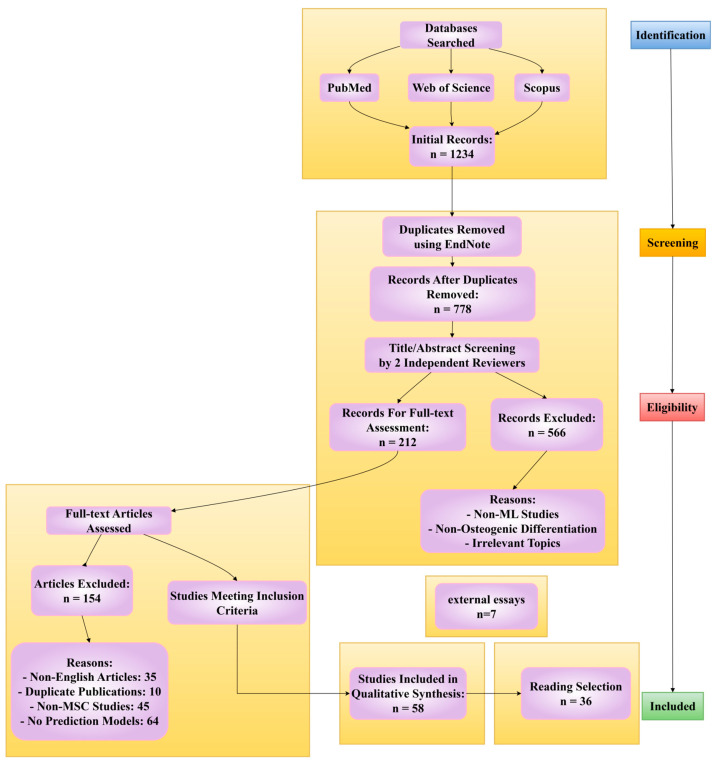
PRISMA Flow Diagram; the total number of cited references is 44.

**Table 1 bioengineering-12-01089-t001:** Comparison of Machine Learning Models for Predicting MSC Osteogenic Differentiation.

Model	Principle	Advantages	Limitations	Application Scenarios	Performance Metrics (References)
ResNet-50	Alleviates gradient vanishing in deep networks via residual blocks and automatically extracts spatial features (e.g., cell edges, textures) from images.	High accuracy (AUC > 0.96); enables early prediction (within 24 h); suitable for high-resolution images.	High computational cost; requires large training datasets; sensitive to imaging parameters.	Morphological image analysis (e.g., live-cell imaging, bright-field images).	AUC > 0.96 (Mai et al.); Accuracy > 96% [8]
LASSO	Uses L1 regularization for feature selection to retain key morphological features (e.g., cell area, shape factor) associated with osteogenic differentiation.	Reduces overfitting; suitable for small sample sizes; non-invasive (avoids cell destruction).	Cannot handle high-dimensional image data; relies on manual feature extraction; lower accuracy than deep learning models.	Morphological parameter analysis (e.g., quantitative cell morphology indices).	Accuracy: 82% [15]
Ridge Regression	Optimizes image acquisition and analysis methods, combining biochemical markers (e.g., ALP activity, calcium deposition) to build prediction models.	Enables early prediction (3-day morphological features predict 3-week results); improves reliability by integrating biochemical markers.	Requires validation with biochemical markers; depends on consistent image acquisition.	Osteogenic prediction combining morphology and biochemical markers.	Accurately predicts 3-week osteogenic differentiation results [19]
Generative Adversarial Network (GAN)	Enhances cell image data through adversarial training of a generator and discriminator, improving model performance in small-sample scenarios.	Solves overfitting in small-sample cases; improves model generalization ability.	Requires high-quality generated data; complex training process.	Small-sample morphological analysis (e.g., limited cell image data).	Accuracy > 85% [22]
Random Walk	random walk on PPI networks to screen core genes.	Uncovers MSC heterogeneity; identifies core osteogenic regulatory genes; constructs gene regulatory networks.	Requires scRNA-seq data; depends on PPI network accuracy.	Transcriptomic data processing (e.g., scRNA-seq analysis).	Identifies osteogenic regulatory genes (e.g., *FOXA1*) [24]
Cross-modal Transformer	Fuses RNA-seq (transcriptomics) and TMT (proteomics) data, capturing time delays between gene expression and protein activity via self-attention mechanisms.	Reveals post-transcriptional regulatory mechanisms; integrates multi-omics data.	Requires multi-omics data; high computational complexity.	Transcriptomics-proteomics integration analysis.	Determines 24 h delay between *ALP* gene expression and protein activity [21]
Support Vector Machine (SVM)	SVM for processing proteomic data to identify differential proteins	Identifies differential proteins in ECM pathways; predicts early metabolic markers.	Requires proteomic/metabolomic data; poor model interpretability.	Proteomic/metabolomic analysis (e.g., data from MSCs of osteoporosis patients).	Identifies 205 differential proteins in ECM pathways (Feng et al.); accuracy: 89% [9]
Random Forest	Integrates multiple decision trees to analyze metabolomic data (e.g., lactate, ATP levels) and correlate metabolites with osteogenic differentiation efficiency.	Resists overfitting; suitable for multi-feature data; identifies early metabolic markers.	Poor model interpretability; long computation time.	Metabolomic data screening (e.g., metabolomic analysis of MSCs in 2D/3D cultures).	Accuracy: 89% [9]

## Data Availability

No new data were created or analyzed in this study.

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
