# Peer review of "Application of Machine Learning in Predicting Osteogenic Differentiation of Mesenchymal Stem Cells"

_bioengineering, 2025, doi:10.3390/bioengineering12101089_

Round 1
Reviewer 1 Report (Previous Reviewer 3)
Comments and Suggestions for Authors
The revised and resubmitted manuscript addressed most of my previous concerns, and I believe the authors' arguments are now more substantiated. I consider the manuscript suitable for publication.
Author Response
Thank you for your reviewing. Your previous suggestions have been extremely helpful and have provided us with valuable insights.
Reviewer 2 Report (New Reviewer)
Comments and Suggestions for Authors
The review article titled "Application of Machine Learning in Predicting Osteogenic Differentiation of Mesenchymal Stem Cells" provides a comprehensive overview of the role of machine learning (ML) in predicting the osteogenic differentiation potential of mesenchymal stem cells (MSCs). The authors discuss three main applications of ML: (1) morphological feature analysis, (2) omics data integration, and (3) drug/biomaterial screening. However, the methodology, interpretation, and presentation of results raise several concerns that undermine the validity and impact of the review findings. Therefore, there are some critical points lead me to reject this review and need to be addressed as follows:
- Page 1 lines 11-24—The abstract needs enhancements. The authors need to first enhance relevance explanation and also provide the main limitations of current study (review) as well as add more findings related to their results in their review. The case that the abstract only provided general information without any deep explanation or adding main findings to the fields.
- The authors overlooked numerous existing review studies on in Predicting Osteogenic Differentiation of Mesenchymal Stem Cells. A simple search would reveal multiple publications (e.g., post-2020 works) that address similar challenges more rigorously. This gap indicates insufficient literature review and undermines the claimed novelty of their review by evaluating all the current related studies to their work. Therefore, there is a clear absence of novelty in the presented work and the authors need to review and show all the related works and their main limitations that the work overcome them. The authors are required to explain the novelties in a clear way. More details are required.
- By the end of the introduction, I highly recommend to add a flowchart showing all the steps and points that you have covered in your review.
- the authors must in more scientific way to justify the reasons for the selection studies in their review. The flowchart in Figure 1 need to be explained in a separated subsection with its information. The case that in the introduction, the authors need to show and explain in a clear way the novelties of their work in comparison with others published work.
- Every single added work need to be reviewed in a way that it can be seen their main strengths, weaknesses, and disadvantages which by the end of every subsection or big part, a conclusion needs to be added as it is a review paper which show the main direction for the work.
- I think the authors need to add a new section explaining the main issues and significance of Osteogenic Differentiation of Mesenchymal Stem Cells and why we need to integrate the ML models. The case that as it is review, the readers need to understand the topic and why we need the ML to be integrated.
- Statements like "ML will revolutionize MSC therapy" are speculative. What concrete evidence supports this beyond computational predictions?
- The discussion on CNNs for morphology analysis (Section 2) repeats known advantages of ResNet-50 (e.g., residual blocks) without addressing its limitations in small datasets or alternatives like Vision Transformers.
- The omics section (Section 3) lists standard tools (e.g., Seurat, random forests) but ignores cutting-edge approaches (e.g., single-cell latent variable models, spatial transcriptomics integration). More details are required to be discussed and explained.
- Why emphasize accuracy/AUC without discussing precision-recall trade-offs for imbalanced datasets?
- The authors highlight ResNet-50’s high accuracy (>96%) but omit its failure cases. How often do these models fail in real-world scenarios (e.g., noisy imaging data)? This is a review so you have to explain your reasoning and justification to show the readers the main direction for any future works.
- The review cites 34 articles despite identifying 58 eligible ones. Why were the remaining 25 not discussed? This selectivity raises concerns about objectivity. Moreover many reviewed works were published from 2013-2014.
- Most of the figures presented in the manuscript are out of quality. Check if this due to the conversion to PDF.
- Pages 8-10 lines 275-331, the conclusion and perspective is also too long and needs improvements, as a review work, it is recommended to have two separate section name as recommendations for future works or something related and then conclusion which provides main findings that explain your findings, not add general findings or describe your work. Moreover, the authors must briefly provide the main limitations of their works and the future directions. Please, read more related reviews to your topic and then understand how to re-structure the review in more interesting and scientific way.
- The manuscript needs to be carefully proofread to improve the overall quality of text, since there are some minor punctuation and grammatical mistakes.
The manuscript needs to be carefully proofread to improve the overall quality of the text, since there are some minor punctuation and grammatical mistakes.
Author Response
Please see the attachment.请见附件。

Reviewer 3 Report (New Reviewer)
Comments and Suggestions for Authors
The review “Application of Machine Learning in Predicting Osteogenic Differentiation of Mesenchymal Stem Cells” by Hanyue Mao, Zheng Zhou, Ying Yang, Kunlu Lin, Chuyao Zhou, and Xiaoyan Wang addresses the multiple applications of machine learning for predicting the osteogenic differentiation of MSCs. It summarizes current approaches, including the analysis of cellular morphological features, omics data, and the screening of drugs and biomaterials, and looks ahead to future directions. The review is highly interesting and innovative; however, it could be further improved:
In the proposed networks, how is the role of the ZNF521 gene, which has recently been suggested to prevent osteoblastogenesis, considered or positioned?
The authors could strengthen the manuscript by discussing the impact of different MSC sources. For example, MSCs derived from Killian nasal polyps display higher differentiation potential into osteocytes and adipocytes compared with MSCs from common nasal polyps. Could the presented models clarify such source-related differences in osteogenic differentiation? The authors might also consider adding a table to emphasize this aspect.
3.In addition, the manuscript could be further enhanced by including a graphical abstract.
Round 2
Reviewer 2 Report (New Reviewer)
Comments and Suggestions for Authors
The authors have partially addressed the comments. However, I have a few more major suggestions as follows:
1. First of all, I encourage the authors to read more related review articles to enhance their understanding of the topic. Additionally, the authors are required to address all my previous comments, as most of them were only partially addressed. I have rejected the paper, and I am unsure why we are still reviewing this article when our comments have not been fully considered.
2. The authors must in more scientific way to justify the reasons for the selection studies in their review. Systematic reviews and meta-analyses should be included to justify the rationale behind selecting these works for the review.
3. The work flowchart in Figure 1 is different from Systematic reviews and meta-analyses so the authors need to show what they have in their study and then for Systematic reviews and meta-analyses, they will explain the justifications for article selections.
4. The authors need to ensure that the size and type of text are consistent across all figures. Please check the flowchart for any discrepancies in text sizes and formatting.
5. Please double-check the grammar throughout the manuscript.
6. I kindly request that these comments be addressed thoroughly.
7. I leave the final decision for the editor for article acceptance.
Author Response
Please see the attachment.请见附件。

Reviewer 3 Report (New Reviewer)
Comments and Suggestions for Authors
The authors significantly improved their manuscript so it is now publishable in bioingegnering journal.
Author Response
Thank you for your reviewing. Your previous suggestions have been extremely helpful and have provided us with valuable insights.谢谢您的审阅。您之前的建议非常有帮助,为我们提供了宝贵的见解。
This manuscript is a resubmission of an earlier submission. The following is a list of the peer review reports and author responses from that submission.
Round 1
Reviewer 1 Report
Comments and Suggestions for Authors
The article is solid in terms of content, but some sections could be improved in terms of clarity, methodology, and interpretation of the results. Adding figures would be helpful to make the topic clearer and more explanatory.
Reviewer 2 Report
Comments and Suggestions for Authors
This is a review of the application of machine learning in predicting osteogenic differentiation of MSCs. Unfortunately, the review is not scientifically sound, it does not present interesting information. The writing style is more like a report not a review! The authors present no critical analysis of the literature, there are no diagrams or figures or tables! MSCs are now termed mesenchymal stromal cells not stem cells.
Comments on the Quality of English LanguageThere are multiple grammatical and structural issues. For example, line 14-17 of the abstract.
Reviewer 3 Report
Comments and Suggestions for Authors
There are some typing errors, such as: Page 1, line 16, “factors, However” (comma instead of dot); page 3, line 101, “factors.s”, among others. These and other similar mistakes should be carefully revised to ensure clarity and consistency in the text.
The manuscript does not provide the methodology used to produce the literature review. No information is provided regarding the search strategy, inclusion/exclusion criteria, databases used, or the rationale behind the selected studies. The incorporation of a flowchart or PRISMA-style diagram to illustrate the selection and screening process would enhance the transparency and reproducibility of the review.
There’s a lack of citations in the development of the manuscript, e.g., in the 1st paragraph of Section 2: “Early Prediction Based on Morphology”, the authors present several assertions and technical descriptions without any supporting references. Given the descriptive nature of a review article, all statements should be accompanied by relevant citations from the literature.
Section 3 discusses the use of omics data but lacks technical depth. Similarly to the previous topic, the authors mention transcriptomics, proteomics, and metabolomics without detailing the specific analytical pipelines, tools, or ML models used in each context. Again, the inclusion of schematic figures illustrating the representative pipelines, or tables summarizing key findings, would significantly enhance the clarity and usefulness of this section.
Since the main topic of the manuscript is the use of Machine Learning in Prediction Models for Osteogenic Differentiation of MSC, the author should provide a more in-depth discussion of their findings. The manuscript mentions various machine learning approaches (e.g., ResNet, LASSO, CNNs), however, it fails to offer a critical comparative analysis of these models. The content would benefit significantly from a more thorough exploration of the principles underlying each algorithm, their respective strengths and limitations, and the contexts in which they have demonstrated their performance. Additionally, metrics such as accuracy, sensitivity, and specificity of the prediction models should be presented to support a more objective evaluation.
The stated objective in the abstract is to "review the progress made in applying machine learning to predict the osteogenic differentiation of mesenchymal stem cells." However, this objective is not explicitly reiterated or expanded upon in the main body of the manuscript. Moreover, the concluding remarks (Section 5) are rather general and do not critically synthesize the findings nor clearly reflect the aim set forth in the abstract. A more cohesive and reflective conclusion is recommended, providing the current gaps, limitations, and prospective research directions in the manuscript’s domain.
While the manuscript summarizes numerous applications of machine learning in the field, it does not offer a unifying perspective or a novel conceptual framework that could advance the current understanding of computational modeling in osteogenic differentiation. A more critical discussion connecting the reviewed approaches to underlying trends, challenges, and methodological advances in machine learning itself—such as model interpretability, generalizability, and data integration—would significantly strengthen the manuscript’s contribution.
RECOMMENDATION:
The manuscript addresses a highly relevant and contemporary topic in regenerative medicine, specifically the application of machine learning techniques to predict osteogenic differentiation in mesenchymal stem cells. While the subject is timely and of considerable interest to the bioengineering community, the manuscript requires substantial revisions to improve its methodological transparency, analytical depth, and structural coherence. Therefore, unfortunately, I do not consider the manuscript suitable for publication at this time. I suggest that the authors revise the manuscript thoroughly and resubmit it for future consideration.